# Biomethanation of Carbon Monoxide by Hyperthermophilic Artificial Archaeal Co-Cultures

**Aaron Zipperle** [1,†] , **Barbara Reischl** [1,2,†] , **Tilman Schmider** [1] , **Michael Stadlbauer** [1] , **Ivan Kushkevych** [3] , **Christian Pruckner** [1] , **Monika Vítězová** [3] and **Simon K.-M. R. Rittmann** [1,2,*]

1   Archaea Physiology & Biotechnology Group, Department of Functional and Evolutionary Ecology, Universität Wien, 1030 Wien, Austria; aaronzipperle@gmail.com (A.Z.); barbara.reischl@univie.ac.at (B.R.); tilman.schmider@uit.no (T.S.); a01005948@unet.univie.ac.at (M.S.); greeese@gmail.com (C.P.)
2   Arkeon GmbH, 3430 Tulln an der Donau, Austria
3   Department of Experimental Biology, Faculty of Science, Masaryk University, 62500 Brno, Czech Republic; kushkevych@mail.muni.cz (I.K.); vitezova@sci.muni.cz (M.V.)
*   Correspondence: simon.rittmann@univie.ac.at; Tel.: +43-4277-76513; Fax: +43-4277-876513
†   These authors contributed equally to this work.

**Abstract:** Climate neutral and sustainable energy sources will play a key role in future energy production. Biomethanation by gas to gas conversion of flue gases is one option with regard to renewable energy production. Here, we performed the conversion of synthetic carbon monoxide (CO)-containing flue gases to methane ($CH_4$) by artificial hyperthermophilic archaeal co-cultures, consisting of *Thermococcus onnurineus* and *Methanocaldococcus jannaschii*, *Methanocaldococcus vulcanius*, or *Methanocaldococcus villosus*. Experiments using both chemically defined and complex media were performed in closed batch setups. Up to 10 mol% $CH_4$ was produced by converting pure CO or synthetic CO-containing industrial waste gases at a high rate using a co-culture of *T. onnurineus* and *M. villosus*. These findings are a proof of principle and advance the fields of Archaea Biotechnology, artificial microbial ecosystem design and engineering, industrial waste-gas recycling, and biomethanation.

**Keywords:** Archaea Biotechnology; anaerobic microbiology; methanogenesis; biohydrogen; biological gas conversion



## 1. Introduction

In the European Union, around 70% of primary energy is generated by the combustion of fossil fuels, contributing about 78% (3367 Tg-$CO_2$ equ.) of the total emitted greenhouse gases [1,2]. A transition to a carbon dioxide ($CO_2$)-neutral and sustainable energy production system is urgently needed. One of the possibilities is to utilize the power-to-gas process [3–5]. Within this process, biomethanation of $CO_2$ to methane ($CH_4$) offers a sustainable opportunity to enable the transition from fossil fuels, as it is an autobiocatalytic process. Therefore, it is envisioned that biomethanation will become an essential part of future energy production systems, as $CH_4$ could be produced at a stable pace and stored in vast amounts in the natural gas grid network [6]. Pure cultures of methanogenic archaea (methanogens) [7–12] and enrichment cultures containing methanogens [13–17] can be utilized for in situ or ex situ biomethanation [18–20].

Carbon monoxide (CO)-containing rich waste gases are a by-product of industrial processes such as steelmaking [21]. CO-containing gases can also be obtained by gasification of carbon-rich materials, such as domestic organic waste or lignocellulose conversion to syngas [22]. The fact that biofuel production directly from lignocellulose is still costly and biotechnologically challenging makes microbial gas to gas conversion from CO a promising alternative bioprocess [7,23]. Developments in the transformation of gaseous waste products to energetically valuable compounds emphasize the potential for syngas as a substrate [24–27]. CO has a high potential for donating electrons, making it a favourable

substrate for lithotrophic microorganisms [28,29]. Besides CO, waste and syngas mainly consist of molecular hydrogen ($H_2$), $CO_2$, and $CH_4$. As early as 1990, it was demonstrated that methanogens can metabolize some components from syngas [30].

The direct conversion of pure CO to $CH_4$ was performed by hydrogenotrophic methanogens and subsequently analysed [31–34]. Furthermore, growth adaption to CO did not change the $CH_4$ production rates significantly in the case of *Methanothermobacter marburgensis*, and the specific growth rate (μ) of *Methanothermobacter thermautotrophicus* on pure CO was only a hundredth of the growth rate achieved using $H_2$:$CO_2$ as a substrate [31]. This led to the assumption that an artificial co-culture, where CO is converted to $CH_4$ in a successive bioprocess, would lead to higher efficiency, as microorganisms specifically adapted to the task of converting CO and producing $CH_4$ can be selected. Studies showed that the independent performance of the water gas shift reaction (WGSR) and biomethanation by two different organisms in the same vessel resulted in a more than 20-fold faster conversion than direct conversion by a single organism [34]. Therefore, we hypothesized that an artificial co-culture of a carboxydotrophic, hydrogenogenic microbe with a hydrogenotrophic, autotrophic organism would drive favourable thermodynamic conditions for the WGSR. These conditions are created by direct removal of the gaseous metabolic end products of the WGSR, that is $H_2$:$CO_2$, by the methanogen. The conversions that are successively performed by the two organisms can be summarized with the following equation:

$$4\,CO + 4\,H_2O \rightarrow 4\,CO_2 + 4\,H_2 \rightarrow CH_4 + 3\,CO_2 + 2\,H_2O \tag{1}$$

Overall, studies using a co-culture approach showed promising results for biomethane production rates [34,35]. Based on a previous study where a bacterial/archaeal co-culture was utilized [36], we wanted to investigate the potential of artificial archaeal co-cultures consisting of a carboxydotrophic, hydrogenogenic archaeon, and different hydrogenotrophic, autotrophic, and methanogenic archaea for biomethanation. We selected *Thermococcus onnurineus* for performing the catalysis of the WGSR, as it was shown that μ and $H_2$ productivity on CO was substantially higher compared to other carboxydotrophic and hydrogenogenic microbes [37–40]. To catalyse the second part of the reaction, *Methanocaldococcus jannaschii*, *Methanocaldococcus vulcanius,* and *Methanocaldococcus villosus* were selected, because of their similar cultivation requirements to *T. onnurineus* with respect to temperature, salt concentration, and pH optimum. All four organisms were isolated from deep-sea hydrothermal vents and belong to the Euryarchaeota. They are able to grow in a temperature range of 63 to 86 °C, a salt concentration of 1 to 5%, and a pH of 5.5 to 7.0. *T. onnurineus* is a heterotroph, while the methanogens are chemolithoautotrophs [41–44]. Moreover, hyperthermophilic organisms are more advantageous over mesophiles, since at higher temperatures, a faster conversion of CO by the carboxydotrophic microorganism occurs, and a three times faster removal of $H_2$ is obtained by the methanogen [34,45]. Therefore, the properties of a hyperthermophilic environment positively affects growth and conversion rates and is, thus, advantageous over mesophilic conditions. Here, we analysed whether *T. onnurineus* together with one of the three methanogens can be grown as a powerful artificial archaeal co-culture to efficiently generate biological $CH_4$ from synthetic waste gases.

## 2. Materials and Methods

### 2.1. Chemicals

CO (99.999 Vol.-%), $H_2$:CO (60 Vol.-% in CO), $H_2$:$CO_2$ (80 Vol.-% in $CO_2$), and an artificial CO-containing syngas ($CO_2$ 16.7 Vol.-%, $H_2$ Vol.-% 16.8%, $CH_4$ Vol.-%14.7, $N_2$ Vol.-% 14.5%, and CO 37.3 Vol.-%) were used for closed batch experiments. For gas chromatography (GC), $H_2$ (99.999 Vol.-%), $CO_2$ (99.999 Vol.-%), CO (99.999 Vol.-%), $H_2/CO_2$ (80 Vol.-% in $CO_2$), $H_2/N_2$ (4.5 Vol.-% $H_2$ in $N_2$), $CH_4$ (99.995 Vol.-%), and the standard test gas (Messer GmbH, Wien, Austria) (containing 0.01 Vol.-% $CH_4$, 0.08 Vol.-% $CO_2$ in $N_2$) were used in addition to the gases mentioned above. All gases, except the standard

test gas, were purchased from Air Liquide (Air Liquide GmbH, Schwechat, Austria). All other chemicals were of the highest grade available.

### 2.2. Media

Medium A is a modified version of the Deutsche Sammlung von Mikroorganismen und Zellkulturen (DSMZ) medium 282. The exact composition of medium A and medium B can be found in Supplementary Materials, Tables S1 and S2. Balch's vitamin solution was used [46]. For *M. marburgensis* and *M. thermautotrophicus*, a phosphate-buffered medium was used [47].

### 2.3. Strains and Cultivation Conditions

The strains *M. jannaschii* JAL-1, *M. villosus* KIN24-T80, *M. vulcanius* M7, *M. marburgensis* DSM 2133 (Marburg), and *M. thermautotrophicus* DSM 1053 (delta H) were purchased from the DSMZ. *T. onnurineus* NA1 was provided by Prof. Dr. Sung Gyun Kang (Korea Institute of Ocean Science and Technology (KIOST), Ansan, Korea).

Every cultivation was performed in closed batch mode [48]. Experiments were conducted in 120 mL serum bottles (Ochs Glasgeraetebau, Langerwehe, Germany), sealed with a 20 mm butyl rubber stopper and an aluminium crimp cap (Chemglass Life Science LLC, Vineland, NJ, USA). Serum bottles were filled with the corresponding medium and sealed. Anaerobic conditions were created by evacuating and re-pressurizing with $H_2:CO_2$ (4:1) to 0.5 barg five times. The bottles were autoclaved and stored at 4 °C until further use. Unless otherwise stated, the medium was augmented with the following sterile filtered stock solutions before inoculation: $NaHCO_3$, L-Cysteine-HCL, and Balch's Vitamin solution [46] (see Supplementary Materials, Tables S1 and S2). Afterwards, the bottles were flushed with $H_2:CO_2$ (4:1) to 0.5 barg, before the addition of autoclaved $Na_2S \cdot 9H_2O$.

The inoculum, 1 mL (2% v/v) of an actively grown culture, was added anaerobically. The final liquid phase in each serum bottle added up to 50 mL. Depending on the experiment, the headspace gas phase was exchanged with the corresponding gas. Bottles were incubated at 80 °C in either a double-layer shaking incubator at 100 rpm (LABWIT ZWYR-2102C, Labwit Scientific Pty Ltd., Burwood East, Australia) or in a water bath at ~100 lateral shakes per minute (GFL 1083, LAUDA-GFL, Burgwedel, Germany).

### 2.4. Pure-Culture Closed Batch Experiments

The methanogens *M. villosus*, *M. vulcanius*, and *M. jannaschii* were grown under 2 barg $H_2:CO_2$ in either medium A or B. A reduced version of them without vitamins, yeast extract, trace elements, or a combination of the three was also tested. The incubation rhythm consisted of 13 and 7 h incubation periods, with 2 h of sampling in between every period. The same rhythm was applied for cultivation of *T. onnurineus*. It was grown in the same medium, but under 1 barg CO. These incubation periods did not apply for *M. marburgensis* and *M. thermautotrophicus*, because of their slow growth. Furthermore, they were grown in a phosphate-buffered *M. marburgensis* medium with either 1.7 barg CO, 1.7 barg $H_2:CO$ (4:1), or 1.7 $H_2:CO_2$ (4:1).

### 2.5. Co-Culture Closed Batch Experiments

The co-cultures consisted of *T. onnurineus* and either *M. villosus*, *M. vulcanius*, or *M. jannaschii*. The experiments were conducted according to two general schemes. In the first setup, the experiment started by growing the respective *Methanocaldococcus* strain under 2 barg $H_2:CO_2$ first. After 13 h in the incubator, *T. onnurineus* was added. The bottles were then pressurized with CO, $H_2:CO$, or artificial syngas to 1 barg. In the second setup, *T. onnurineus* grew 13 h under pure CO, $H_2:CO$, or artificial syngas before one of the *Methanocaldococcus* strains was added. From this point onward, both schemes followed the same incubation rhythm, as described in the pure-culture experiments, and ended after a maximum of 80 h of cumulative incubation time. Every experiment was performed in hexuplicates (n = 6) on media A and B, as well as on reduced versions of them.

## 2.6. Sampling

The routinely performed sampling consisted of removing the cultures from the incubator and measuring the pressure as soon as the bottles cooled down to room temperature (~60 min). To analyse growth, 0.7 mL of the cultures were withdrawn for optical density (OD) measurements ($\lambda$ = 578 nm). Lastly, the bottles were flushed and re-pressurized with the corresponding gas.

## 2.7. Analytical Procedures

OD of the cultures was measured via a spectrophotometer at $\lambda$ = 578 nm (Specord 200 Plus, Analytik Jena, Jena, Germany). The headspace gas composition before and after inoculation in pure cultures was determined via the gas headspace pressure difference [18,47,49]. Gas evolution and uptake rates were calculated according to methods described in refs. [47,50,51]. Samples that were withdrawn for gas chromatography (GC) analysis did not undergo the sampling procedure. The headspace gas composition in co-culture experiments was analysed via GC, and the evolution and uptake rates were calculated [52]. Some of the negative controls revealed "air contamination" and were removed from the calculations of the results. To maintain the correct atmosphere, OD was measured after completion of the GC run and as such is only an estimate of the true OD.

## 3. Results

### 3.1. Growth Kinetics of Methanogens in Defined Medium

Growth and gas conversion by the methanogens *M. villosus*, *M. vulcanius*, and *M. jannaschii* were analysed in defined versions of media A and B on $H_2:CO_2$ (4:1), by removing complex components. Removal of the trace elements from the media hindered growth. *M. villosus* showed a higher methane evolution rate (MER) than the other two methanogens in medium A and B (Figure 1).

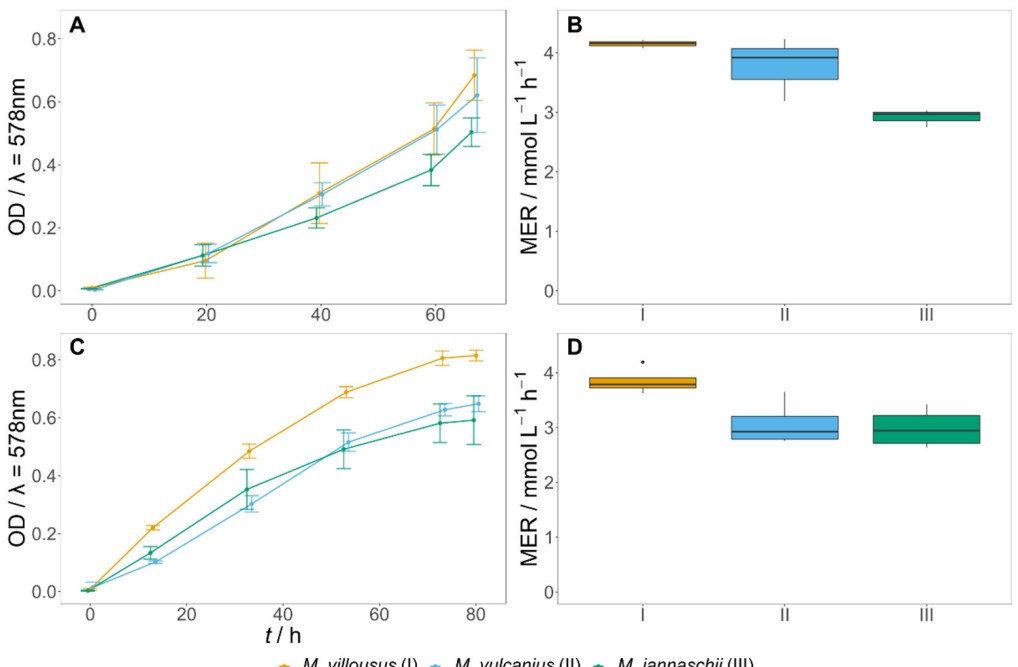

**Figure 1.** Growth kinetics and MER *of M. villosus, M. vulcanius*, and *M. jannaschii.* (**A,B**) grown in medium A; (**C,D**) grown in medium B. Both media were without yeast extract and vitamins. Error bars in the line graphs show the standard deviation. All experiments are N = 1, n = 3. However, N = 2, n = 3 for *M. villosus* using medium B.

*M. marburgensis* and *M. thermautotrophicus* were grown on CO, $H_2$:CO (3:1) and $H_2$:$CO_2$ (4:1) [53]. The analysis showed that *M. marburgensis* and *M. thermautotrophicus*, when grown on $H_2$:$CO_2$ (4:1), had a high turnover, and furthermore, experiments on $H_2$:CO resulted in poor $CH_4$ conversion. No growth was observed on pure CO (see Supplementary Materials, Figure S1, Table S5). Due to these findings, *M. marburgensis* and *M. thermautotrophicus* were excluded from the co-culture experiments.

### 3.2. T. onnurineus Grown on CO

*T. onnurineus* was grown individually in a pure culture in media A and B. Results indicated that the organism reached higher $OD_{578}$ when grown on medium A as well as achieved a higher gas conversion than in media B. Omission of vitamins from medium B showed no difference regarding the $H_2$ evolution rate (HER) (see Supplementary Materials, Table S6, Figure S2).

### 3.3. Artificial Archaeal Co-Culture Engineering

Conversion of CO to $CH_4$ and $CO_2$ was performed with a co-culture consisting of *T. onnurineus* together with either *M. villosus*, *M. jannaschii*, or *M. vulcanius*. After pre-growth of the methanogens for 13 h, the cultures were inoculated with *T. onnurineus*. Growth and gas rates were analysed for all three co-cultures in media A and B (Tables 1 and 2, Figure 2). Co-cultures in medium A had relatively similar MERs ranging from 1.4 to 1.6 mmol $L^{-1}$ $h^{-1}$, whereas in medium B, the co-culture consisting of *M. villosus* and *T. onnurineus* achieved MERs between 1.6 and 2.0 mmol $L^{-1}$ $h^{-1}$. In general, the measured mean gas evolution and uptake rates were either equal or slightly higher in medium B. The higher CO uptake rate (COUR) and $CO_2$ evolution rate (CER) in medium B indicated that *T. onnurineus* performed better under this condition. It is likely that the $H_2$ uptake rate (HUR) of the methanogens positively influenced the COUR of *T. onnurineus* by creating favourable thermodynamic conditions [34].

**Table 1.** Mean gas evolution and uptake rates of the co-cultures cultivated in medium A on 100% CO [1].

| Co-Culture | MER/mmol $L^{-1}$ $h^{-1}$ | HUR/mmol $L^{-1}$ $h^{-1}$ | CUR/mmol $L^{-1}$ $h^{-1}$ | COUR/mmol $L^{-1}$ $h^{-1}$ | CER/mmol $L^{-1}$ $h^{-1}$ |
|---|---|---|---|---|---|
| *M. villosus* + *T. onnurineus* | 1.4 ± 0.4 | 6.4 ± 1.5 | 1.4 ± 0.4 | 6.4 ± 1.5 | 5.0 ± 1.1 |
| *M. vulcanius* + *T. onnurineus* | 1.6 ± 0.3 | 7.0 ± 1.3 | 1.6 ± 0.3 | 7.0 ± 1.3 | 5.4 ± 0.9 |
| *M. jannaschii* + *T. onnurineus* | 1.5 ± 0.4 | 6.5 ± 1.5 | 1.5 ± 0.3 | 6.5 ± 1.5 | 5.0 ±1.2 |

[1] Measurements were taken after a 7 h incubation period. Data collected at indicated timepoints in Figure 2 A. (N = 1, n = 6). Values are shown with standard deviation.

**Table 2.** Mean gas evolution and uptake rates of the co-cultures cultivated in medium B on 100% CO [1].

| Co-Culture | MER/mmol $L^{-1}$ $h^{-1}$ | HUR/mmol $L^{-1}$ $h^{-1}$ | CUR/mmol $L^{-1}$ $h^{-1}$ | COUR/mmol $L^{-1}$ $h^{-1}$ | CER/mmol $L^{-1}$ $h^{-1}$ |
|---|---|---|---|---|---|
| *M. villosus* + *T. onnurineus* | 2.0 ± 0.3 | 8.3 ± 1.3 | 2.1 ± 0.3 | 8.4 ± 1.3 | 6.3 ± 0.9 |
| *M. vulcanius* + *T. onnurineus* | 1.6 ± 0.5 | 10.3 ± 0.2 | 2.6 ± 0.3 | 11.3 ± 0.7 | 8.7 ± 0.5 |
| *M. jannaschii* + *T. onnurineus* | 1.8 ± 0.4 | 9.0 ± 1.5 | 1.9 ± 0.5 | 9.1 ± 1.6 | 7.2 ± 1.2 |

[1] Measurements were taken after a 7 h incubation period. Data collected at indicated timepoints in Figure 2 B. (N = 1, n = 6). Values are shown with standard deviation.

The results suggested that the co-culture with *M. vulcanius* was less efficient in medium B, as it had a higher $CO_2$ uptake rate (CUR) and HUR, compared to medium A, but not a higher MER. This agrees with the results obtained from pure-culture experiments, where medium B led to a slightly lower MER of *M. vulcanius* compared to the other methanogens.

Comparison of the MERs between the pure and the co-cultures suggested that, on average, a lower MER was measured in the co-cultures (Tables 1 and 2, Figure 1). A likely reason for this was that they were dependent on the conversion of CO to $CO_2$ and $H_2$ by *T. onnurineus*. Figure 3 shows that the $CH_4$ production and growth occurred at the highest capacity, as the $H_2$ content in most cultures was below 1 mol%, except for one case, suggesting it was fully converted. As such, the rate-limiting step in $CH_4$ production by the methanogens in the co-culture was the availability of $H_2$.

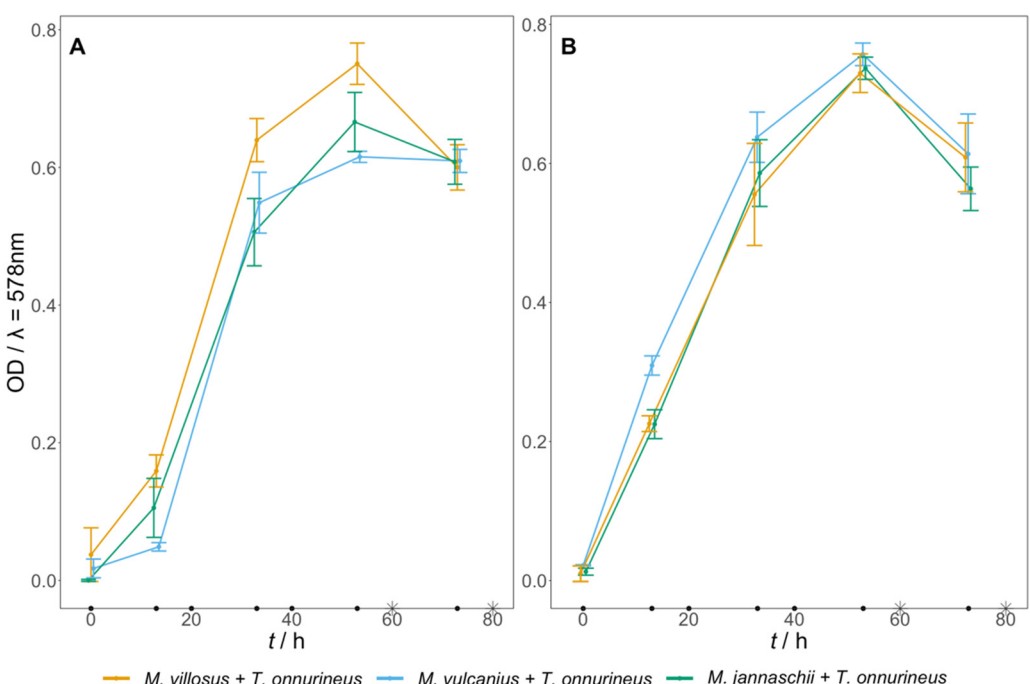

**Figure 2.** Growth kinetics of either *M. villosus*, *M. jannaschii*, or *M. vulcanius* with *T. onnurineus*. After the first 13 h, the medium was inoculated with *T. onnurineus* and the gas phase was exchanged from $H_2:CO_2$ to CO. The error bars show the standard deviation. Closed black circles on the *x*-axis mark the re-pressurising of the headspace with CO. The asterisks on the *x*-axis indicate the sampling points for GC analysis. (**A**) Growth in medium A; (**B**) growth in medium B. N = 1, n = 6.

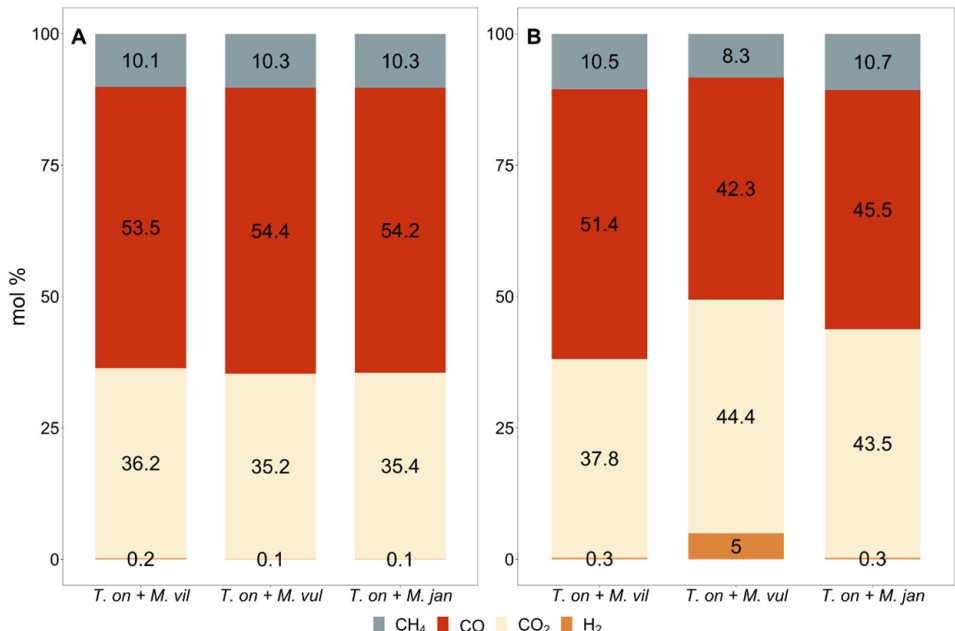

**Figure 3.** Relative mean molar gas composition in the co-cultures headspace after a 7 h incubation period on CO. Measurements were taken at the timepoints indicated in Figure 2. (**A**) Co-culture cultivated in medium A; (**B**) co-culture cultivated in medium B. N = 1, n = 6.

To circumvent $H_2$ limitation in the co-culture experiments, two strategies were employed. First, to create a functional culture with a high $H_2$ production rate in advance of the methanogen being added, an inoculation scheme was tested, where *T. onnurineus* was initially grown 13 h on CO, prior to addition of said methanogen. Second, a different gas

composition, $H_2$:CO (3:1) instead of pure CO, was used to provide additional $H_2$ available in the atmosphere. In addition, an experiment with a gas composition resembling that of industrial waste or syngases was performed [21]. The co-cultures were exposed to an artificial syngas (see Section 2.1) that could arise by industrial processes (Table 3). As *T. onnurineus* grown with *M. villosus* in medium B was considered the best co-culture, this pair was used. However, it should be noted that these were closed batch experiments and that the productivities are dependent on the conversion kinetics and on the tested time interval.

**Table 3.** Mean gas evolution rates of co-cultures in medium B under different gas compositions and inoculation orders [1].

| Co-Culture | Gas | MER/mmol $L^{-1}$ $h^{-1}$ | CER/mmol $L^{-1}$ $h^{-1}$ |
|---|---|---|---|
| *M. villosus* + *T. onnurineus* | CO | $2.0 \pm 0.3$ | $6.3 \pm 0.9$ |
| *T. onnurineus* + *M. villosus* | CO | $1.1 \pm 0.1$ | $3.7 \pm 1.0$ |
| *M. villosus* + *T. onnurineus* | $H_2$:CO | $1.0 \pm 0.4$ | $0.0 \pm 0.0$ |
| *T. onnurineus* + *M. villosus* | $H_2$:CO | $1.0 \pm 0.3$ | $0.0 \pm 0.0$ |
| *M. villosus* + *T. onnurineus* | Art. syngas | $1.5 \pm 0.5$ | $0.9 \pm 0.6$ |
| *T. onnurineus* + *M. villosus* | Art. syngas | $0.9 \pm 0.3$ | NA |

[1] Measurements were taken after a 7 h incubation period (N = 1, n = 6). Values are shown with standard deviation.

In experiments with pure CO, the MER was twice as high when *M. villosus* was added first. The average $H_2$ content turned out to remain below 0.3 mol% (see Supplementary Materials, Table S7). When gassing with $H_2$:CO, interestingly, both inoculation schemes showed a decreasing performance over time, as revealed by a MER, which was only half of what was achieved when pure CO was used (Table 3). However, the measured $CO_2$ levels were below 0.1 mol% in both cases (see Supplementary Materials, Table S7). This could have been caused by the presence of $H_2$ in the $H_2$:CO, as the additional application of $H_2$ within the gas negatively affected growth and performance of *T. onnurineus*, which in turn most likely hindered the MER. The reduced CER by *T. onnurineus* may have been caused by the lower partial pressure of CO in the serum bottle headspace.

In the experiment where the artificial syngas mix was used and where *M. villosus* was pre-grown, the MER was $1.5 \pm 0.5$ mmol $L^{-1}$ $h^{-1}$. The average $CH_4$ mol% increased by about 10% to 24.8 mol%. This increase was similar to what was obtained when using pure CO, although the MER decreased as the experiment progressed. We did not observe a negative impact on the MER during co-culture experiments when elemental sulphur was added.

## 4. Discussion

This study is a new brick in the emerging research field of Archaea Biotechnology [54] and artificial microbial ecosystem design and engineering [55]. We investigated the conversion of one-carbon substrates (CO, $H_2$:CO, and CO-rich waste gases) by the artificially designed co-cultures of either *M. villosus*, *M. jannaschii*, or *M. vulcanius* together with *T. onnurineus*. Our results showed fast and reliable gas conversion, with a reduction in pure CO to about 50 mol% and simultaneous production of ~10 mol% $CH_4$ within 7 h. The most efficient conversion of the artificial syngas was performed by the co-culture *M. villosus* with *T. onnurineus*, inoculated in this order. This culture showed a CO reduction of 7%, starting from 37.3 mol% and an increase in $CH_4$ by ~10 mol% within 7 h. This proved the ability of the co-cultures to convert a variety of different CO-containing gas compositions with a lower proportion of CO.

A study with a similar experimental setup reported an ~6% $CH_4$ increase after 22 h [35]. We obtained 10% increase in 7 h. This suggests that the established co-culture (*T. onnurineus/M. villosus)* is of higher catalytic power. Nonetheless, most up-to-date published co-cultures were tested in different setups than the one reported in this study, making a direct comparison of evolution and uptake rates rather difficult [26,30,34]. Establishing different co-cultures in a bioreactor setup will, thus, be of great importance to fully understand their growth and production kinetics.

Pure-culture closed batch experiments of the methanogens in a defined medium on $H_2$:$CO_2$ showed a higher MER ($4.2 \pm 0.1$ mmol $L^{-1}$ $h^{-1}$) than the co-cultures ($2.0 \pm 0.3$ mmol $L^{-1}$ $h^{-1}$) (Figure 1, Table 3). Therefore, the potential for achieving a higher MER in co-cultures is possible if the necessary gas supply can be performed, and the inhibitory concentration of CO would not be surpassed [56]. Although the main limitation of the co-culture grown on pure CO was the availability of $H_2$ for methanogenesis and, hence, the conversions of CO to $H_2$ and $CO_2$ by *T. onnurineus* (Figure 3 and Supplementary Materials, Table S7). The addition of $H_2$ to the gas phase did not provide an increase in the MER. Rather, it led to a limitation of $CO_2$ availability (Supplementary Materials, Table S7), most likely due to the reduced performance of *T. onnurineus* under the lower CO partial pressure. Consequently, application of the artificial syngas led to an increase in MER, as biomethanation seems to be neither limited by $CO_2$ nor $H_2$ availability. Nonetheless, it did not reach the same values as in pure CO (Table 3). This can be explained by the concentration reduction in CO, $CO_2$, and $H_2$ in syngas by the other initially present gases, reducing substrate availability.

A change in the cultivation method to a bioreactor setup with higher pressure, agitation, and a constant gassing, resulting in having a higher gas solubility and higher gas transfer rate to the liquid phase, might increase the conversion of CO by *T. onnurineus* and the potential of the co-culture [37,39,48]. However, the ratio of the gas in the liquid phase has to be considered carefully, as high agitation might also lead to an excessive CO availability for the methanogens [34]. As recently more and more genetic tools for archaea become available, an overexpression of the carbon monoxide dehydrogenase could also be a solution to debottleneck CO conversion by *T. onnurineus* [39,57].

Unfortunately, *T. onnurineus* is still dependent on a complex medium containing yeast extract, which limits the potential industrial applicability. However, co-cultivation could be used to gain advantages or improve growth of both microorganisms through improving their syntrophic relationships [34]. Finding a suitable defined medium for *T. onnurineus* or replacing it by a different organism that can catalyse the WGSR from CO in minimal conditions is one suggested avenue of research. *Carboxydocella thermautotrophica* or *Carboxydocella sporoproducens* would fall into the same pH and temperature conditions as the herein employed methanogens [35].

A direct conversion of CO to $CH_4$ by a single organism such as *M. marburgensis* or *M. thermautotrophicus* is likely not the most suitable biotechnological approach, as the inhibitory concentration of CO for these methanogens is very low [31,34]. The artificially created archaeal co-cultures consisting of one of the hyperthermophilic methanogenic archaea *M. villosus*, *M. jannaschii*, and *M. vulcanius* together with *T. onnurineus*, as performed in this study, are highly efficient and reliable for biomethanation. By adaptation to a minimal medium and by performing targeted bioprocess development, artificial archaeal co-cultures could be of environmental, economic, and industrial value in renewable energy production and storage.

## 5. Conclusions

This contribution is another step in the rapidly developing research field of Archaea Biotechnology. Furthermore, it is a proof of principle for the design and engineering of artificial microbial co-cultures. Additionally, it marks a cornerstone in the use of artificial archaeal co-cultures for the biomethanation of syngas. This design of artificial archaeal co-cultures represents a novelty in the field of biomethanation, due the unique selection of

organisms and their hyperthermophilic growth conditions. The created co-cultures convert a variety of different CO-containing gases, including syngas, to $CH_4$. As the WGSR might still be the limiting factor, finding the best organism for conversion of $H_2O$:CO to $H_2$:$CO_2$ will be of future relevance to enhance the kinetics of the artificial co-culture. Apart from the choice of the archaea, the inoculation sequence of the strains and medium development are crucial factors for a successful design. These factors will be important in engineering the next steps for scaling up the bioprocess of artificial archaeal co-cultures. Inferring from our results, hyperthermophilic and anaerobic archaea are of great biotechnological relevance, as they act as highly efficient $H_2$ and $CH_4$ cell factories in artificial co-culture.

**Supplementary Materials:** The following are available online at https://www.mdpi.com/article/10.3390/fermentation7040276/s1, Table S1: Chemical composition of medium A, Table S2: Chemical composition of medium B, Table S3: 141 trace element solution modified from DSMZ, Table S4: Holden's trace element solution 2, Table S5: The physiological maximal and mean values of $CH_4$ production and growth kinetics of *M. marburgensis* and *M. thermautotrophicus*, Table S6: HER of *T. onnurineus* after 7 h of incubation, Table S7: Relative mean molar gas composition of the co-culture's headspace after a 7 h incubation period, Figure S1: Growth of *M. marburgensis* and *M. thermautotrophicus* in defined medium, Figure S2: Growth of *T. onnurineus* at 1 barg CO.

**Author Contributions:** Conceptualization, A.Z., T.S. and S.K.-M.R.R.; methodology, B.R., I.K., M.V. and S.K.-M.R.R.; validation, A.Z., B.R. and S.K.-M.R.R.; formal analysis, A.Z.; investigation, A.Z., B.R., T.S., M.S. and C.P.; resources, M.V. and S.K.-M.R.R.; data curation, A.Z. and B.R.; writing—original draft preparation, A.Z., B.R. and S.K.-M.R.R.; writing—review and editing, A.Z., B.R., I.K., M.V. and S.K.-M.R.R.; visualization, A.Z.; supervision, S.K.-M.R.R.; project administration, M.V. and S.K.-M.R.R.; funding acquisition, M.V. and S.K.-M.R.R. All authors have read and agreed to the published version of the manuscript.

**Funding:** Greatly acknowledged is the Österreichische Forschungsförderungsgesellschaft (FFG) for funding the project BioHyMe (grant no. 853615). The BMBWF is acknowledged for supporting the research with the WTZ project CZ 08/2020. Open access funding by the University of Vienna.

**Institutional Review Board Statement:** Not applicable.

**Informed Consent Statement:** Not applicable.

**Data Availability Statement:** The datasets used and/or analysed during the current study are available from the corresponding author on reasonable request.

**Acknowledgments:** S.K.-M.R.R. greatly acknowledges Sung Gyun Kang (Korea Institute of Ocean Science and Technology (KIOST), Ansan, Reublic of Korea) for providing *T. onnurineus* NA1. We want to thank Angus Hilts for comments on and proofreading of the manuscript.

**Conflicts of Interest:** B.R. and S.K.-M.R.R. declare competing interests due to their employment in the Arkeon GmbH. All other authors declare no competing interest.

## Abbreviations

| | |
|---|---|
| $CO_2$ | Carbon dioxide |
| $CH_4$ | Methane |
| CO | Carbon monoxide |
| $H_2$ | Molecular hydrogen |
| WGSR | Water gas shift reaction |
| DSMZ | Deutsche Sammlung von Mikroorganismen und Zellkulturen |
| OD | Optical density |
| GC | Gas chromatography |
| MER | Methane evolution rate |
| HER | Molecular hydrogen evolution rate |
| COUR | Carbon monoxide uptake rate |
| CER | Carbon dioxide evolution rate |
| HUR | Molecular hydrogen uptake rate |
| CUR | Carbon dioxide uptake rate |

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
