# Peer review of "Biomethanation of Carbon Monoxide by Hyperthermophilic Artificial Archaeal Co-Cultures"

_fermentation, doi:10.3390/fermentation7040276_

Round 1

Reviewer 1 Report

The current work established a co-system combing bacteria and archaea for CO conversion to methane, which is an interesting work. However, according to my reviewing, there was already similar work already (i.e. High rate biomethanation of carbon monoxide-rich gases via a thermophilic synthetic coculture). So, what is the main novelty of the present work compared with the already submitted papers? Is the conversion rate higher or ……..? So, the author should pay more attention to the statement of the novelty of the work.

The author also stated that yeast extract also should added in the medium for better growth of the microorganism. According to my experience and my work, it is better to investigate the auxotroph of bacteria (for example, adding specific amino acid maybe could increase the growth of bacteria due to that some bacteria could not synthesize some important amino acid and …….). it will also benefit the pilot-scale or industrialization scale for syngas biomethanation in pure culture.

Even the author stated the bacteria and archaea had the similar growth environment, I think the bacteria and archaea had different niche for growth, which indicated that separate the Bactria and archaea with different niche might showed better co and syngas conversion performance.

In addition, the author should add more discussion especially comparison with other research work. And the conclusion also need improvement.

Author Response

Reviewer 1

The current work established a co-system combing bacteria and archaea for CO conversion to methane, which is an interesting work. However, according to my reviewing, there was already similar work already (i.e. High rate biomethanation of carbon monoxide-rich gases via a thermophilic synthetic coculture). So, what is the main novelty of the present work compared with the already submitted papers? Is the conversion rate higher or ……..? So, the author should pay more attention to the statement of the novelty of the work.

Answer:

Thank you for the review. Yes, Diender et al 2018 had conducted some important fundamental research closely related to our work. The novelty here is that this is the first study done only with a co-culture of archaea, no bacteria were involved this setup. Further, growth and conversion was performed under hyperthermophilic conditions, which was also never used for this application before. We tired to emphasize these novelties more in the Title, Discussion and Conclusion.

The author also stated that yeast extract also should added in the medium for better growth of the microorganism. According to my experience and my work, it is better to investigate the auxotroph of bacteria (for example, adding specific amino acid maybe could increase the growth of bacteria due to that some bacteria could not synthesize some important amino acid and …….). it will also benefit the pilot-scale or industrialization scale for syngas biomethanation in pure culture.

Answer:

Thank you for this comment. To date, Thermococcus onnurineus is yet still depended on yeast extract for growth. We agree, as also stated in this manuscript, that using yeast extract limits the potential industrial applicability and finding a suitable media is a crucial step for further studies.

Even the author stated the bacteria and archaea had the similar growth environment, I think the bacteria and archaea had different niche for growth, which indicated that separate the Bactria and archaea with different niche might showed better co and syngas conversion performance.

Answer:

Thank you for this comment. It is an important contribution. It is likely possible that cultivation in separate optimized conditions would lead to better growth, than when both organisms are grown in the same condition and medium. This is especially the case for the methanogens, as they are under severe stress with the used high concentrations of CO. Nevertheless, the catalysis of the water gas shift reaction (WGSR) is the limiting step in syngas methanation and not the methanation.

The WGSR can be enhanced by creating thermodynamically favourable conditions, which happens when the products (H2 and CO2) are directly removed. This direct removal is maximised when the methanogen is as close as possible to the organism catalysing the WGSR.

That’s where the power of the co-culture lies. Growing both in the same vessel creates a superior conversion than by separating the reactions/organisms. We tried to further emphasize this concept in the Discussion.

In addition, the author should add more discussion especially comparison with other research work. And the conclusion also need improvement.

Answer:

Thank you for this comment. We rearranged and extended the discussion and the conclusion. Comparison to Kohlmayer et al. 2018 was done, which further emphasized the potential of the used co-culture. This was implemented in the discussion. Comparing to similar studies like Diender et al. is unfortunately not possible as we worked in a close batch system. Most other research was conducted in bioreactors under constant gas exchange and thus not comparable to our results. Therefore, we further emphasised that testing our co-culture in a bioreactor setup will be of importance to be able to compare to other publications. Moreover, we tied to improve the conclusion by greater implementation of the important knowledge that was gained from the experiments and by including future prospective.

Reviewer 2 Report

Paper is well written, however, language check is required to make more meaningful.

Quality of figures need to be improved 

conclusion must be more meaning full with some future perspective 

Latest references must be cited form good journals like Fermentation.

Author Response

Reviewer 2

Paper is well written, however, language check is required to make more meaningful.

Answer:

Thank you for reviewing our manuscript. A language check was performed as requested. The manuscript was now proef read for grammar and spelling mistakes by a native speaking scientist (see Acknowledgment).

Quality of figures need to be improved

Answer:

Thank you for this comment. We enlarged the legend and lines in figure 2 for better visualisation and hope to have fulfilled your requirements. More precise suggestions are highly appreciated.

Conclusion must be more meaning full with some future perspective

Answer:

Thank you for this comment. We implemented more of our novelty and milestones, which also includes the limitations of the system. Proceeding from there, future perspectives were given. We hope that the colusions read now very well.

Latest references must be cited form good journals like Fermentation.

Answer:

Thank you for this comment. We included following 4 relevant works that were published in Fermentation.

  1. de Medeiros et al. Multi-Objective Sustainability Optimization of Biomass Residues to Ethanol via Gasification and Syngas Fermentation: Trade-Offs between Profitability, Energy Efficiency, and Carbon Emissions. Fermentation 2021

  2. Phillips et al. Syngas Fermentation: A Microbial Conversion Process of Gaseous Substrates to Various Products. Fermentation 2017

  3. Westman et al. Syngas Biomethanation in a Semi-Continuous Reverse Membrane Bioreactor (RMBR). Fermentation 2016

  4. Devarapalli et al. Continuous Ethanol Production from Synthesis Gas by Clostridium Ragsdalei in a Trickle-Bed Reactor. Fermentation 2017

Reviewer 3 Report

Report of the manuscript “Biomethanation of carbon monoxide by artificial archaeal co-cultures” by Zipperle et al.

First of all, I would like to state that this is a very interesting paper and deserves to be published!

Still, I have a few remarks.

The concept is very interesting; while archaea-based methanization is more and more popular and has already gained a lot of attention in the field of Energy Sustainability, methanization of CO has less attention. Obviously, there are fewer CO-source than CO2, but concerning the nature of CO, its removal and methanization should be very important.

Since some of the readers will be less biology- or chemistry-oriented (but rather energy-engineers, interested in the proper handling of flue-gas), I would like to make two recommendations.

  • It would be helpful to give some citation for CO2 methanization together with 1-2 sentences. CO2 methanization is more known for energy engineers, and from that few sentences, they might be able to position CO methanization better. A good citation would be a very recent review of Prof. Sterner: Power-to-Gas and Power-to-X—The History and Results of Developing a New Storage Concept, Energies 2021, 14(20), 6594; https://doi.org/10.3390/en14206594
  • It would be useful to give a list of abbreviations. For a microbiologist, seeing OD (Figures 1,2) is clear, but energy engineers might need some source to know that it is optical density. Even the Authors might write a sentence that optical density is a measure of the archaea-concentration.

With these changes, the field of potential readers will be much broader.

There is one more problem, namely the length of the Conclusion, which is even shorter than the abstract. I would recommend to expand this part; six lines is definitely suitable as Conclusion.

Due to the necessary major rewriting of the Conclusions, I would recommend major revision for the manuscript.

Author Response

Reviewer 3

Report of the manuscript “Biomethanation of carbon monoxide by artificial archaeal co-cultures” by Zipperle et al.

First of all, I would like to state that this is a very interesting paper and deserves to be published!

Answer:

Thank you for the nice words and for reviewing our manuscript.

Still, I have a few remarks.

The concept is very interesting; while archaea-based methanization is more and more popular and has already gained a lot of attention in the field of Energy Sustainability, methanization of CO has less attention. Obviously, there are fewer CO-source than CO2, but concerning the nature of CO, its removal and methanization should be very important.

Since some of the readers will be less biology- or chemistry-oriented (but rather energy-engineers, interested in the proper handling of flue-gas), I would like to make two recommendations.

It would be helpful to give some citation for CO2 methanization together with 1-2 sentences. CO2 methanization is more known for energy engineers, and from that few sentences, they might be able to position CO methanization better. A good citation would be a very recent review of Prof. Sterner: Power-to-Gas and Power-to-X—The History and Results of Developing a New Storage Concept, Energies 2021, 14(20), 6594; https://doi.org/10.3390/en14206594

Answer:

Thank you for this important suggestion. We changed our Introduction accordingly and hope to have managed to point out the importance of methanation from CO.

It would be useful to give a list of abbreviations. For a microbiologist, seeing OD (Figures 1,2) is clear, but energy engineers might need some source to know that it is optical density. Even the Authors might write a sentence that optical density is a measure of the archaea-concentration.

Answer:

Thank you for this comment. A list of abbreviations was added at the end of the manuscript.

In the section Sampling, we extended a sentence to explain that the optical density is used for growth analysis.

With these changes, the field of potential readers will be much broader.

There is one more problem, namely the length of the Conclusion, which is even shorter than the abstract. I would recommend to expand this part; six lines is definitely suitable as Conclusion.

Answer:

Thank you for this comment. We revised and extended our conclusion, by implementing important findings of the study and limitations of the co-culture. This lead to an increase in length and hopefully also in relevance of the conclusion.

Due to the necessary major rewriting of the Conclusions, I would recommend major revision for the manuscript.

Reviewer 4 Report

The article is devoted to solving an important problem - the utilization of CO with the formation of such biofuels as methane. This is a good work describing the research methods and results in sufficient details. In the Discussion Section, the authors summarize the data obtained and formulate the tasks for further research.

Minor comments

Lines 48-49 Methanothermobacter marburgensis and the growth rate of M. thermautotrophicus

Lines 71-72 Methanocaldococcus jannaschii, M. vulcanius and M. villosus

Lines 290-291 Carboxydocella thermautotrophica or C. sporoprducens

Table 3 H2:CO

However, the list of references is drawn up extremely carelessly with numerous inaccuracies. Also in the references, about 30 % of references related to the studies of the authors; I don’t think this is acceptable.

Author Response

Reviewer 4

The article is devoted to solving an important problem - the utilization of CO with the formation of such biofuels as methane. This is a good work describing the research methods and results in sufficient details. In the Discussion Section, the authors summarize the data obtained and formulate the tasks for further research.

Answer:

Thank you for the review and the following comments.

Minor comments

Lines 48-49 Methanothermobacter marburgensis and the growth rate of M. thermautotrophicus

Lines 71-72 Methanocaldococcus jannaschii, M. vulcanius and M. villosus

Lines 290-291 Carboxydocella thermautotrophica or C. sporoprducens

Table 3 H2:CO

Answer:

Thank you for these precise comments. We changed the text accordingly.

However, the list of references is drawn up extremely carelessly with numerous inaccuracies. Also in the references, about 30 % of references related to the studies of the authors; I don’t think this is acceptable.

Answer:

Thank you for stating that error. We manually curated and corrected the references accordingly. Also, within the text non-essential references were removed and 5 new references from different groups were added.

Round 2

Reviewer 1 Report

The author have well answered mu questions and the manuscript was well improved, which can be accepted in the present form.